# Direct observation of ultrafast symmetry reduction during internal conversion of 2-thiouracil using Coulomb explosion imaging

Till Jahnke [1,2] ✉, Sebastian Mai [3], Surjendu Bhattacharyya [4], Keyu Chen [4], Rebecca Boll [2], Maria Elena Castellani[5], Simon Dold [2], Ulrike Frühling[6], Alice E. Green [2,7], Markus Ilchen[2,6,8], Rebecca Ingle [9], Gregor Kastirke[10], Huynh Van Sa Lam [4], Fabiano Lever[6], Dennis Mayer[6], Tommaso Mazza [2], Terence Mullins [2], Yevheniy Ovcharenko [2], Björn Senfftleben [2], Florian Trinter [11], Atia-Tul-Noor [6], Sergey Usenko [2], Anbu Selvam Venkatachalam[4], Artem Rudenko [4], Daniel Rolles [4], Michael Meyer [2], Heide Ibrahim[12,13] & Markus Gühr [6,14] ✉

The photochemistry of heterocyclic molecules plays a decisive role for processes and applications like DNA photo-protection from UV damage and organic photocatalysis. The photochemical reactivity of heterocycles is determined by the redistribution of photoenergy into electronic and nuclear degrees of freedom, initially involving ultrafast internal conversion. Most heterocycles are planar in their ground state and internal conversion requires symmetry breaking. To lower the symmetry, the molecule must undergo an out-of-plane motion, which has not yet been observed directly. Here we show using the example of 2-thiouracil, how Coulomb explosion imaging can be utilized to extract comprehensive information on this molecular deformation, linking the extracted deplanarization of the molecular geometry to the previously studied temporal evolution of its electronic properties. Particularly, the protons of the exploded molecule are well-suited messengers carrying rich information on its geometry at distinct times after electronic excitation. We expect that our new analysis approach centered on these peripheral protons can be adapted as a general concept for future time-resolved studies of complex molecules in the gas phase.

The photochemistry of heterocyclic compounds is crucial for processes and applications such as UV lesions in DNA[1], organic photocatalysis[2], and light harvesting[3]. The photochemical reactivity is determined by the redistribution of photoenergy into electronic and nuclear degrees of freedom. This redistribution involves ultrafast nonradiative processes including internal conversion and intersystem crossings, which are very efficient yet also extremely complex due to their intrinsic electron-nuclear coupling mechanisms, which encompass the concept of conical intersections[4,5]. The challenge in understanding these nonradiative processes in heterocyclic compounds has spurred significant research efforts[6–9]. A fundamental understanding of these processes holds promise for

the rational design of heterocyclic compounds for specific applications.

The internal conversion in heterocyclic molecules involves nuclear motion. Heterocyclic molecules generally absorb light via a strong $^1\pi \to \pi^*$ transition as shown in Fig. 1a. From this $^1\pi\pi^*$ state, the lower-lying $^1n\pi^*$ state can often be reached via internal conversion. The $^1n\pi^*$ state takes the role of a 'doorway state' for intersystem crossing to lower-lying $^3\pi\pi^*$ states. These $^3\pi\pi^*$ states cannot be efficiently populated directly from the $^1\pi\pi^*$ state due to El-Sayed's rule[10]. The photo-excited $^1\pi\pi^*$ state and the doorway $^1n\pi^*$ state belong to different irreducible representations of the $C_s$ point group of planar heterocycles, which are a prominent molecular class. The $^1\pi\pi^*$ and $^1n\pi^*$ states belong to A' and A" irreducible representations, respectively. The discussed A' → A" internal conversion formally requires the activation of at least one out-of-plane molecular motion belonging to the A" irreducible representation, i.e., internal conversion necessitates some out-of-plane molecular motion. The out-of-plane motion for 2-thiouracil, the molecule in our study, is indicated in Fig. 1c. Generally, the out-of-plane motion determines the speed and thus the efficiency of the internal conversion and with it the fate of the photoinduced reactivity. Measuring the ultrafast change of the molecular geometry without environment disturbance recently became possible by diffraction techniques in the gas phase[11] and has been successfully applied to heterocycles[12]. While these techniques retrieve changes in internuclear separations, they are not adept at detecting symmetry changes. In our present study, we fill this gap in the understanding of heterocycles by establishing time-resolved X-ray-induced Coulomb explosion imaging (CEI) employing hydrogen atoms as messengers for the molecule's symmetry.

We choose the case of the thionated nucleobase 2-thiouracil (2-tUra, see scheme in Fig. 1b) for our study. Thionated nucleobases show an efficient population of long-lived triplet states after ultraviolet

excitation[13,14]. This efficient intersystem crossing is a hallmark of this class of molecules and has important consequences for applications in photodynamic therapy and photo-crosslinking studies[14–16]. Some thiobases are used as medicinal drugs (e.g., for immunosuppression[17]) and can induce serious light-triggered damage in patients due to their triplet-state-induced photochemistry. A detailed understanding of the triplet-state chemistry is thus crucial to explore potential applications. Among the different thionucleobases, the microscopic origin of the efficient relaxation into the triplet state is best investigated in 2-tUra. Several simulations point at a mechanism that mostly occurs via the $^1n\pi^*$ state as a doorway state out of the optically excited $^1\pi\pi^*$ state[18,19]. These simulations also suggest that the out-of-plane molecular motion (indicated in Fig. 1c) induces the internal conversion from $^1\pi\pi^*$ to $^1n\pi^*$. In 2-tUra, the out-of-plane motion from a planar symmetry (point 1 in Fig. 1c) is a characteristic local pyramidalization at the $C_2$ atom (see point 2 in Fig. 1c). This motion is accompanied by an elongation of the C-S bond. The pyramidalization at the $C_2$ atom is suggested to happen within the first 100 fs after the UV excitation and guides the molecule from the Franck-Condon region over a $^1\pi\pi^*/n\pi^*$ conical intersection towards the minimum of the $^1n\pi^*$ state[18,19]. Close to the $^1n\pi^*$ minimum (reached after about 100 fs), the sulfur atom is suggested to extend strongly out of the ring plane (point 3 in Fig. 1c) and the molecule resides in this configuration throughout the rest of the relaxation into the triplet states. The changes in the electronic state of the molecule along this pathway were obtained in a previous study[20]. There, Mayer et al. deduced the electronic character from distinct shifts in the X-ray photoelectron spectra, as shown in Fig. 1d. The figure shows a false-color representation of the change in the sulfur 2p photoelectron spectra recorded at different delays after UV excitation. To accomplish this, we plot the difference in the S 2p photoelectron spectra with and without optical excitation as a function of the sulfur 2p binding energy

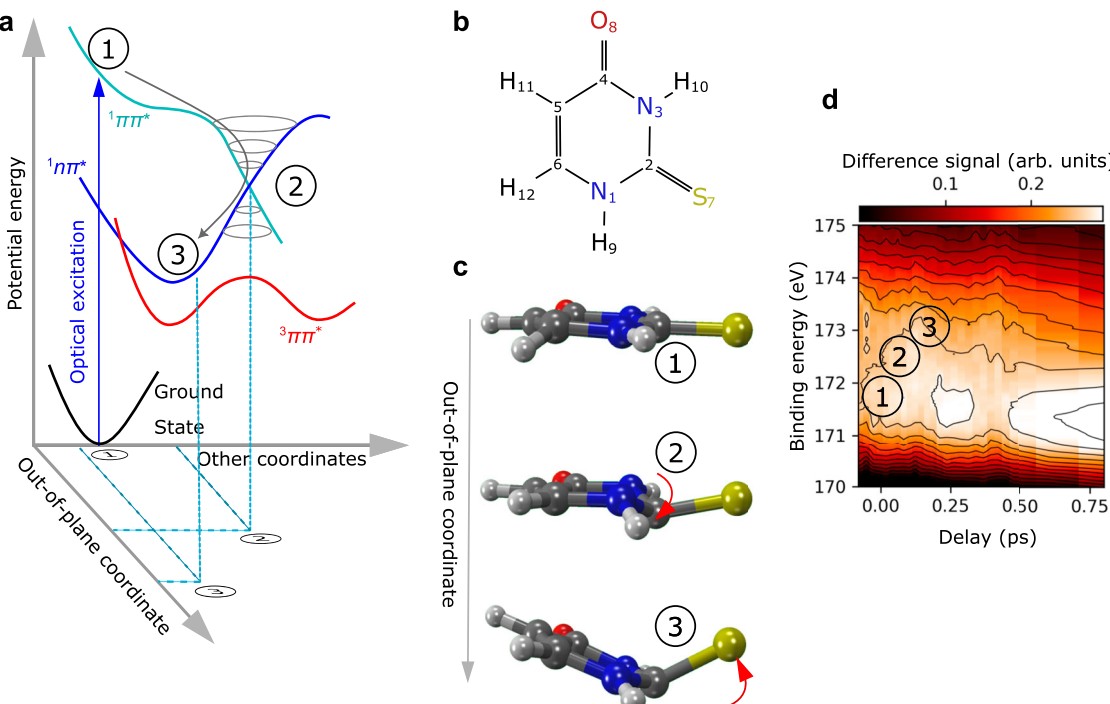

**Fig. 1 | Internal conversion of a heterocyclic molecule via out-of-plane motion.** **a** Sketch of the molecular potential energy surfaces as a function of molecular geometry. States are named by their dominant electronic character. The optical excitation into the $^1\pi\pi^*$ state occurs at the Franck-Condon point 1. The molecule then travels under involvement of the out-of-plane coordinate to point 2. There it undergoes an internal conversion into the $^1n\pi^*$ state via a conical intersection. Around the $^1n\pi^*$ minimum at point 3, the molecule undergoes intersystem crossing into the $^3\pi\pi^*$ state. **b** Schematic of the 2-tUra molecule. **c** Simulated geometries for 2-tUra at

points 1,2, and 3 from ref. 18 with the symmetry-breaking out-of-plane motions being indicated by the red arrows (data used from 'Intersystem Crossing Pathways in the Noncanonical Nucleobase 2-Thiouracil: A Time-Dependent Picture.', by Mai, S., Marquetand, P., and González L., ref. 18, https://doi.org/10.1021/acs.jpclett.6b00616, copyright by American Chemical Society, disclaimer of warranties at https://pubs.acs.org/page/policy/authorchoice_ccby_termsofuse.html, licensed under CC-BY http://creativecommons.org/licenses/by/4.0/). **d** Time-resolved XPS showing a shift in binding energy during the internal conversion (data used from ref. 20).

and the delay. The data were obtained at the FLASH free-electron laser facility at much softer x-ray photon energies as described in Ref. 20. The observed increase in binding energy from point 1 to point 3 resulted from the increase in $^1n\pi^*$ character of the molecule's electronic configuration. This is due to the higher positive local charge at the sulfur atom in the $^1n\pi^*$ state compared to the optically excited $^1\pi\pi^*$ state, which reduces the S $2p$ core-hole screening and thereby increases its binding energy.

In this work, we demonstrate how X-ray-induced CEI monitors the out-of-plane motion of the molecule during the $^1\pi\pi^*$ to $^1n\pi^*$ internal conversion. Furthermore, we directly reference the change in structural symmetry to the aforementioned changes in the electronic structure of the molecule[20]. While CEI itself does not provide direct information about the involved electronic states, we discover that the hydrogen atoms of the molecule are ideal messengers for the molecular symmetry. We plot the momenta which the hydrogen atoms gather after the ionization and Coulomb explosion in a symmetry-adapted reference frame. We deduce the out-of-plane symmetry change from the loss of compactness of the hydrogen features in this frame. In addition, we confirm the symmetry sensitivity of this momentum representation by simulating CEI data with modeled molecular trajectories. The combination of XPS data with the symmetry-sensitive X-ray CEI gives a complete picture of the origin (symmetry) and effect (electronic-character change) in the internal conversion. While we successfully applied the time-resolved X-ray CEI approach to molecular symmetry on one particular heterocyclic molecule, there are no fundamental obstacles that limit its generalization to other heterocycles and even beyond. For instance, the crucial $^1\pi\pi^*/^1\pi\sigma^*$ internal conversion in planar organic systems follows the same symmetry rules as the $^1\pi\pi^*/^1n\pi^*$ states described above[21]. We conclude that this new method for assessing the fundamental symmetry properties during internal conversion will be one major step on the path to new design paradigms for photochemical function.

## Results and discussion

### Experimental conditions for Coulomb explosion imaging

In our present work, we employ time-resolved CEI[22] to examine the out-of-plane nuclear symmetry change in the heterocycle. An initial UV pulse excites the molecule into the $^1\pi\pi^*$ state, and the triggered molecular symmetry change is then probed after a controlled delay by X-ray-induced CEI. The CEI probe technique relies on a rapid charge-up of the molecule that causes (ideally) its instantaneous fragmentation into atomic ions. The full three-dimensional momenta of the atomic fragments are measured in coincidence by a COLTRIMS reaction microscope[23,24], which is a technique that allows to target single molecules in the gas phase. These fragment momenta can then be transformed into a molecular frame of reference[25]. The inversion of the measured momentum-space data to real space is viable for small molecules[26–31]. In most cases, however, such an inversion is not performed, but momentum and energy distributions from the charge-up and CE are modeled and compared to the measured results[32]. CEI has been pioneered several decades ago, using first thin foils[33], and later short IR laser pulses, synchrotrons or (X-ray) free-electron lasers for the charge-up[25,27,29,32,34–45]. In particular, ultrashort/ultraintense pulsed X-rays allow for generation of more than one charge per absorbed photon, which helps to fulfill the requirement of a close to instantaneous charge-up of the molecule.

Our experiment was performed at the Small Quantum Systems (SQS) instrument of the European X-ray Free-Electron Laser (EuXFEL). UV laser pulses (266 nm) were used for electronic excitation of the 2-thiouracil molecules and intense X-ray pulses (hv = 2.6 keV, duration 8–10 fs) were employed for ionization. Please refer to the Methods section for details on the experimental setup, the properties of the UV and free-electron laser light, and the data analysis.

### Definition of coordinate frames and data representation

Figure 2 depicts our momentum-space results in a molecular frame of reference defined by the emission directions of the sulfur and oxygen ions. The measured $S^+$ momentum defines the $z$ axis and spans the $y$-$z$ plane together with the $O^+$ momentum. The momenta of all detected ions are then transformed into this coordinate frame by a rotation and are, in addition, normalized by the magnitude of the $S^+$ momentum. It has been shown that this normalization creates particularly clear momentum-space images when inspecting ring-like molecules[32]. Figure 2a shows the projection of these normalized momenta of the $O^+$ ions and protons on the $y$-$z$ plane. It depicts cleanly separated features corresponding to the four different hydrogen atoms of the molecule. In Fig. 2b, we plot the projection of the normalized proton momenta on the $x$-$z$ plane. Despite being a planar molecule, the protons exhibit a nonnegligible contribution of out-of-plane emission, which probably corresponds to ground- or (thermally) excited-state vibrational motion in the unpumped initial state. The sharp momentum-space features of the emitted protons in Fig. 2 indicate a rapid fragmentation of the molecule. In contrast, the corresponding molecular-frame momenta of the carbon ions are smeared out, as we show in the Methods section. Here, the X-ray-induced charge-up dynamics seem to prohibit an easy extraction of information on properties of the initial molecular geometry without referring to detailed modeling of the ionization process. However, as the protons are typically emitted very rapidly during the initial phase of the X-ray- induced charge-up[32], they still carry the desired information on the molecular symmetry as we show in the following.

Figure 2c depicts the three-dimensional angular emission distribution of the protons in a molecular frame defined by the difference and sum momenta of the $S^+$ and $O^+$ ions (see Methods section for details). The coordinate frame is depicted schematically at the left, including the definitions of the azimuthal and polar angles $\Phi$ and $\Theta$, respectively. The protons labeled 10 and 12 are located at the north and south poles, respectively. Proton 9 is located at an azimuthal angle of $\Phi = 0°$, proton 11 is located at $\Phi = \pm180°$ (wrapping around). The S and O symbols mark the approximate emission direction of the sulfur and oxygen ions (in the planar configuration of the molecule).

### Time-resolved results and simulations: oxygen-sulfur emission angle

After inspecting the static molecular CEI patterns, we now turn to the time-resolved data. In order to observe a UV-induced out-of-plane deformation of the 2-tUra molecule (Fig. 1c), we focus on the relative emission direction of the sulfur and oxygen ions first. Theory predicts that the sulfur atom moves out of the molecular plane from point 1 to 3 in Fig. 1c. This rather drastic change of geometry should be observable already in the relative emission angle between the sulfur and the oxygen ions upon fragmentation by Coulomb explosion. In order to check this assumption, we refer to the modeled geometries from the molecular dynamics (trajectory) dataset, initially presented in refs. 18,19. We emulated the Coulomb explosion on this trajectory data by setting the charge of all atoms of the molecule to +1 at different time steps of the molecular-dynamics simulation and then using our Coulomb explosion code to obtain the final-state momenta of all ions after fragmentation (see Methods section for details). Figure 3a shows the temporal evolution of the cosine of the relative emission angle $\alpha$ between the $S^+$ and $O^+$ ions as predicted by this modeling. The Coulomb explosion results have been smoothed in order to highlight the overall trends despite the discrete nature of the modeled trajectory dataset. At short times after the UV excitation, $\cos(\alpha)$ is strongly peaked at a value of approximately −0.4. The distribution of $\cos(\alpha)$ broadens for larger delays with a more asymmetric contribution towards larger $\cos(\alpha)$ values. In Fig. 3b, we show the measured $\cos(\alpha)$ distribution for unpumped (blue symbols) and UV-pumped molecules with a time delay of $\Delta t = 1000$ fs (red symbols). In line with the

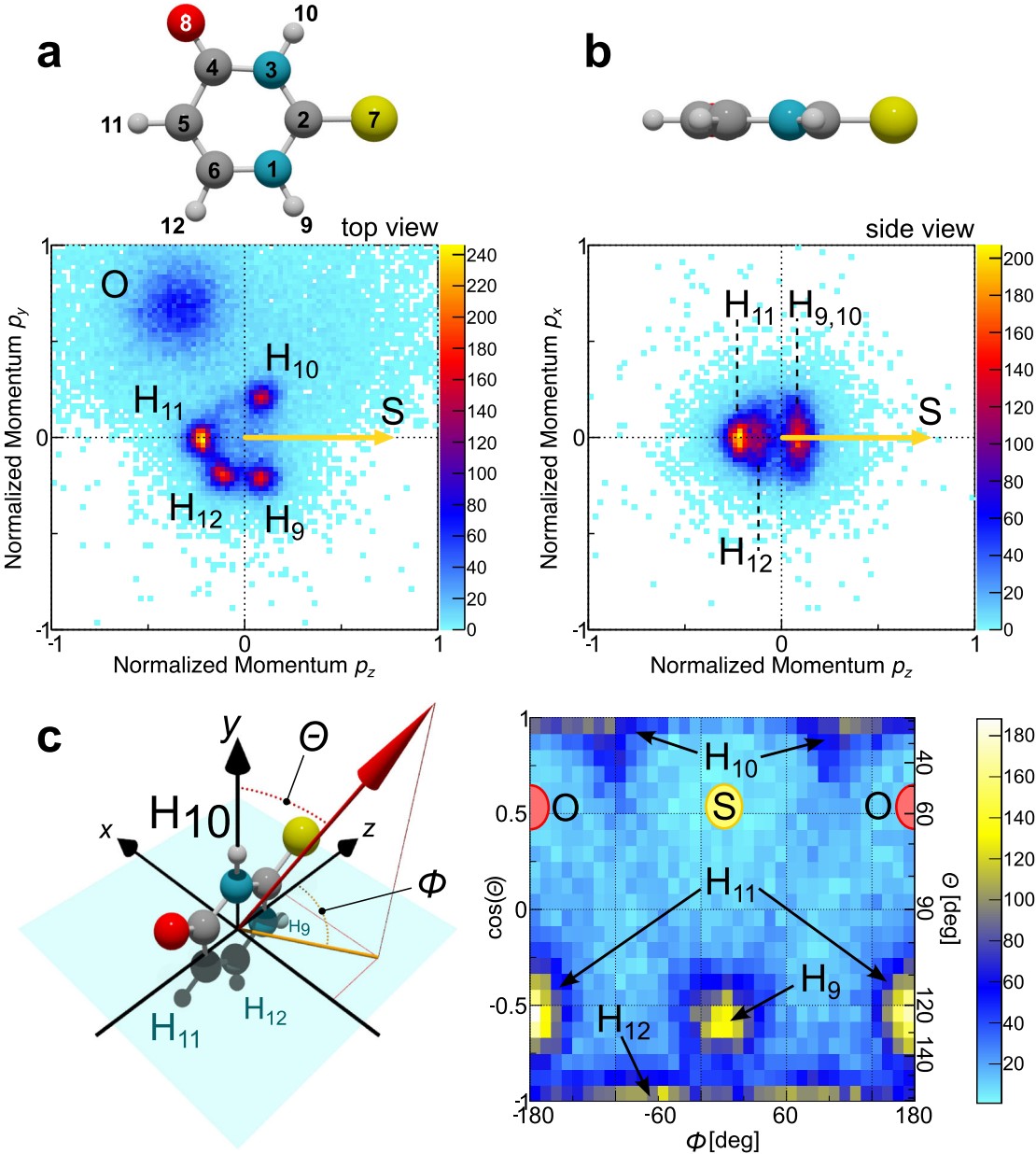

**Fig. 2 | Coulomb explosion momentum-space images of 2-thiouracil without UV excitation.** The molecular coordinate frame is defined by the emission direction of the S atom providing the $z$ axis. The $y$-$z$ plane is spanned by the emission directions of the $S^+$ and $O^+$ ions. The $x$ axis is perpendicular to the $y$-$z$ plane. **a** "Top view" projection of the measured momentum-space data of the oxygen ions and the protons. **b** "Side view" projection of the measured proton momenta. The momenta have been normalized such that the magnitude of the sulfur momentum equals 1 for each molecular ionization event. **c** Angular emission distribution of the protons in the molecular frame indicated by the scheme at the left, which is defined by the difference and sum momenta of the $S^+$ and $O^+$ ions. In the given representation by the spherical coordinates $\Theta$ and $\Phi$ (right panel), $H_{11}$ and O appear at two positions as the $\Phi$ angle wraps around. $H_{10}$ and $H_{12}$ are located at the poles and thus extend across a wide range of $\Phi$. For details see the main text.

modeling, the experimentally determined distribution broadens and contributes to a larger extent on the side of higher $\cos(\alpha)$ values. Panel (c) shows the evolution of $\cos(\alpha)$ as difference plots between the UV-pumped and unpumped cases for four different pump-probe delays ($\Delta t$). The top row shows the results from trajectory modeling, the bottom row presents the experimental data. The measured distributions are wider compared to the ones obtained from theory, but most of the trends in the simulation are present in the experiment, as well. The main negative peak close to $\cos(\alpha) = -0.4$ becomes more pronounced over time, as does the asymmetry of the broadening of the $\cos(\alpha)$ distribution that is observed in Fig. 3a, b. The negative dip in the experiment in Fig. 3c is much broader than in the simulation. We

attribute the discrepancy in this width to an idealized modeling of the Coulomb explosion, not covering a finite charge-up time and neglecting the overall dynamics during the charge-up process. This results in a ground-state distribution in our simulated data which is narrower in $\cos(\alpha)$ than the distribution measured in the experiment. Interestingly, the peak of the measured angle $\alpha$ is very close to the initial angle between the C-O and C-S bonds, which is 116° in the $S_0$ state[19]. Such good mapping from the initial position-space angle to momentum space by the Coulomb explosion is surprising, as already from smaller molecules it is known that there are several properties of a real Coulomb explosion (long-range repulsion, charge distribution not being pointlike), which typically skew the finally observed emission

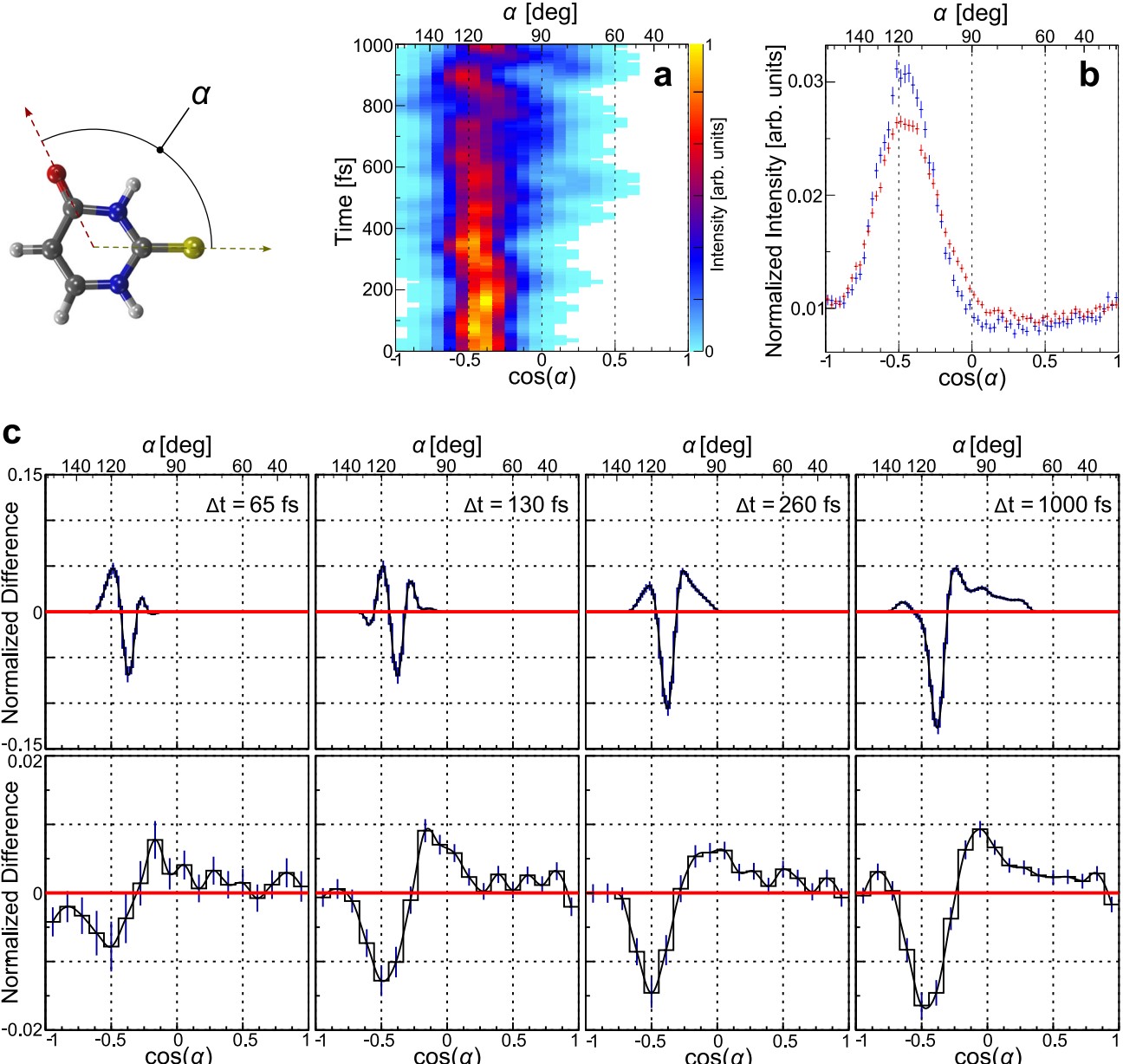

**Fig. 3 | UV-induced changes of the relative emission angle α between the sulfur and the oxygen atoms. a** Temporal evolution of cos(α) predicted by our theoretical modeling. **b** Measured relative emission angle UV-pumped (red) and unpumped (blue). **c** Difference between UV-pumped and unpumped distributions of cos(α) for several time delays between the pump and the probe pulse. Top: modeling, bottom: measured results. The error bars in panels (**b**, **c**) show the standard deviation of the statistical errors of the measurement.

angles[46]. Finally, the angle between the C-O and C-S bonds approaches 104°. This is in line with the increase in intensity in the range of cos(α) = 0.1 for later times in Fig. 3c.

## Time-resolved results and simulations: proton angular distributions

Finally, we arrive at analyzing the out-of-plane motion. To accomplish this, we now inspect the three-dimensional angular emission distributions of the protons shown in Fig. 2c as a function of the delay between the UV and the X-ray pulses. The reasoning behind this approach is twofold. First, if the sulfur moves out of the plane, any atom becomes a reporter since the molecular coordinate frame employed in our observations is spanned by the sulfur and oxygen sum and difference momenta. Second, as the symmetry of the ring changes, the bond directions with the hydrogen change, as well, as depicted by the geometry sketches in Fig. 1c. In the top row of Fig. 4,

we present the corresponding results obtained from our excited-state trajectory-based modeling. Note that the intensity (indicated by the color map) is normalized on total ions per panel to visualize the relative intensity changes between different delay steps. At short times after the UV excitation, the proton emission directions are well-defined in the cos(Θ)-Φ space. As time evolves, firstly, these proton peaks broaden. Secondly, filament-like structures emerge in the range of −0.1 < cos(Θ) < 0.6 marked by the solid-line box. The bottom row depicts our measured results for the same time delays. The contrast is much weaker as compared to the top row, as only a part of the ground-state population in the experiments is UV-excited. Nevertheless, the loss of intensity in the peaks corresponding to $H_9$ (dotted-line frame) and $H_{11}$ (dashed-line frame) is clearly visible, as is an enhancement of the 'filament' structure at larger delays. The presence of traces of these filaments even at the earliest delays, although not as pronounced as at late times, may be due to

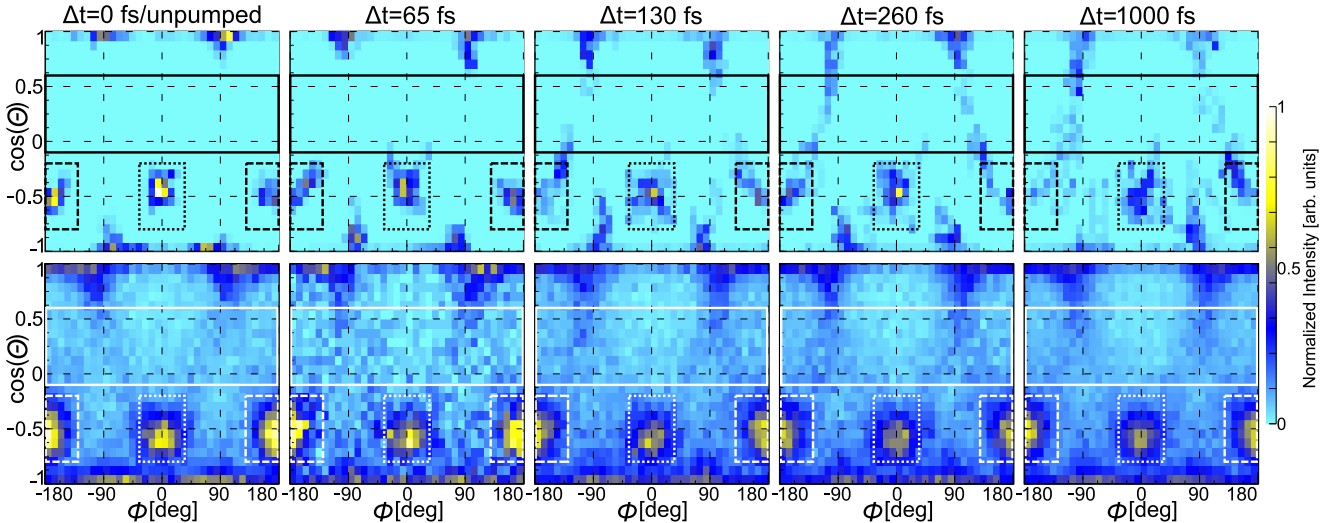

**Fig. 4 | Time-resolved proton emission direction in the spherical coordinates $\Phi$ and $\Theta$.** Progression of the proton angular distributions for different pump-probe delays as provided by our quantum simulations (top row)[19] and measurements (bottom row). The signals at different delays are normalized to the integrated number of ions per delay step. The features associated with $H_9$ are in the dotted-line frame, $H_{11}$ is contained in the dashed-line frames. The so-called filaments are framed by the solid-line box.

vibrationally excited molecules (that are still in their electronic ground state), which occur as the powder sample is heated in the oven source in order to generate the gas-phase target.

We obtain quantitative insight into the out-of-plane motion by integrating over the different areas marked in Fig. 4 by the boxes. Figure 5 shows how these integrals change for different pump-probe delays in theory (line) and experiment (dots with statistical error bars). In Fig. 5a, the integrated intensity belonging to the peak of the $H_9$ atom (dotted-line boxes in Fig. 4) is depicted, which diminishes in both the experiment and the simulation almost instantaneously, i.e., already 65 fs after the UV excitation. In comparison, the change of intensity of the $H_{11}$ peak (dashed-line boxes in Fig. 4, with the integrated signal shown in Fig. 5b) has a delayed onset, with the first decrease setting in at the next measured delay point of 130 fs after the UV pulse. In Fig. 5c, the increase observed in the "filament region" (solid-line boxes in Fig. 4) starts after a slight delay, as well. We interpret these findings in the following way: within 65 fs after the UV excitation, the $H_9$ signal in the indicated dotted-line area diminishes and the angular emission distribution of $H_9$ broadens in the $\Phi$-$\Theta$ plane. We deduce from the experiment that the $H_9$ moves out of the plane spanned by the difference and sum of the oxygen and sulfur momenta, thus lowering the molecular symmetry from planar $C_s$ to nonplanar $C_1$ rapidly. We interpret this as a consequence of the $N_1$-$C_2$-$N_3$ triangle and the attached protons moving out of the molecular plane as indicated by the theoretical prediction for this time step shown in Fig. 5, G2. At larger delays, the diminished $H_9$ signal in the dotted-line frame stems from the sulfur atom moving out of the molecular plane (Fig. 5, G3). In this geometry, all of the hydrogen atoms are located outside of the $y$-$z$ plane, spanned by the oxygen and sulfur sum and difference momenta. Thus, starting at 130 fs after UV excitation, also the $H_{11}$ signal in the dashed-line box diminishes due to the sulfur shifting out of the molecular ring plane. The difference in the onset of the decay of the $H_9$ and $H_{11}$ signals illustrates the differences between the onset of geometries in Fig. 5, G2 and G3.

We now directly compare the results on the out-of-plane nuclear motion from Fig. 5 to the electronic-state measurement via XPS shown in Fig. 1d. The early (at 65 fs) onset of geometry G2 [Fig. 5, also shown in Fig. 1c (2)] is the hallmark of a fast change from a planar to a nonplanar (puckered) geometry. In the XPS data (Fig. 1d) we observe the time-resolved XPS signal to have already shifted by half of its maximal amount at 65 fs, indicating significant $^1n\pi^*$ population. The comparison of the CEI and XPS results at this early time thus directly shows that the

molecular geometry pathway for the ultrafast $^1\pi\pi^*$-$^1n\pi^*$ internal conversion is described by the $C_2$ atom moving out of the plane according to geometry G2. Breaking the planar ($C_s$) symmetry is crucial for the nonadiabatic dynamics. First, the fast deformation of the molecule by a light atom (i.e., the carbon atom $C_2$) determines the path of the molecular wave packet via the $^1\pi\pi^*$-$^1n\pi^*$ conical intersection. Only after that, the mode including the heavier sulfur sets in. This is confirmed by the delayed onset of the corresponding feature in our CEI data, which thus agrees with the intuitive argument that motion of light atoms can induce nonadiabatic dynamics fast and thus efficiently[18].

## Conclusion

Our experiment allowed us to directly monitor the out-of-plane motion of the molecule, which according to theory induces the ultrafast $^1\pi\pi^*$/$^1n\pi^*$ internal conversion. We obtained this direct view into the molecular dynamics using time-resolved X-ray CEI by using the emitted protons (which show well-defined features in momentum space) as messengers for details on the structural deformations of the molecule. We observe the deplanarization of the molecule first at the $C_2$ atom at the earliest time after the excitation, which then continues after a short delay at the S atom. We are able to correlate the geometric properties of the molecule extracted from CEI with the $^1\pi\pi^*$/$^1n\pi^*$ electronic-state changes observed in X-ray photoelectron spectroscopy[19,20]. The combination of both delivers a comprehensive picture of the important degrees of freedom in the coupled electronic and nuclear dynamics of 2-tUra. Theory on other thionucleobases predicts smaller out-of-plane distortions that are much shorter lived, but still accomplish high $^1\pi\pi^*$/$^1n\pi^*$ internal conversion[47]. Further studies using time-resolved X-ray CEI bear the chance for a deeper understanding on how the out-of-plane mode is influenced by molecular geometry via inertial and electronic changes, thus providing potential to actively control the internal conversion of heterocycles. The change of electronic symmetry from A' to A" irreducible representations also holds for the large and important class of $^1\pi\pi^*$/$^1n\pi^*$ internal conversion, playing a major role in amino acid photochemistry. Thus, time-resolved CEI will be important to a large field of molecular processes. Possible candidates need to be prepared in the gas phase, and the size of these molecules is limited by the number of atomic ions that can be generated by single XFEL light pulses. However, with respect to the number of ions that needs to be detected in coincidence, our messenger-atom approach shows that already a threefold ion-coincidence measurement can

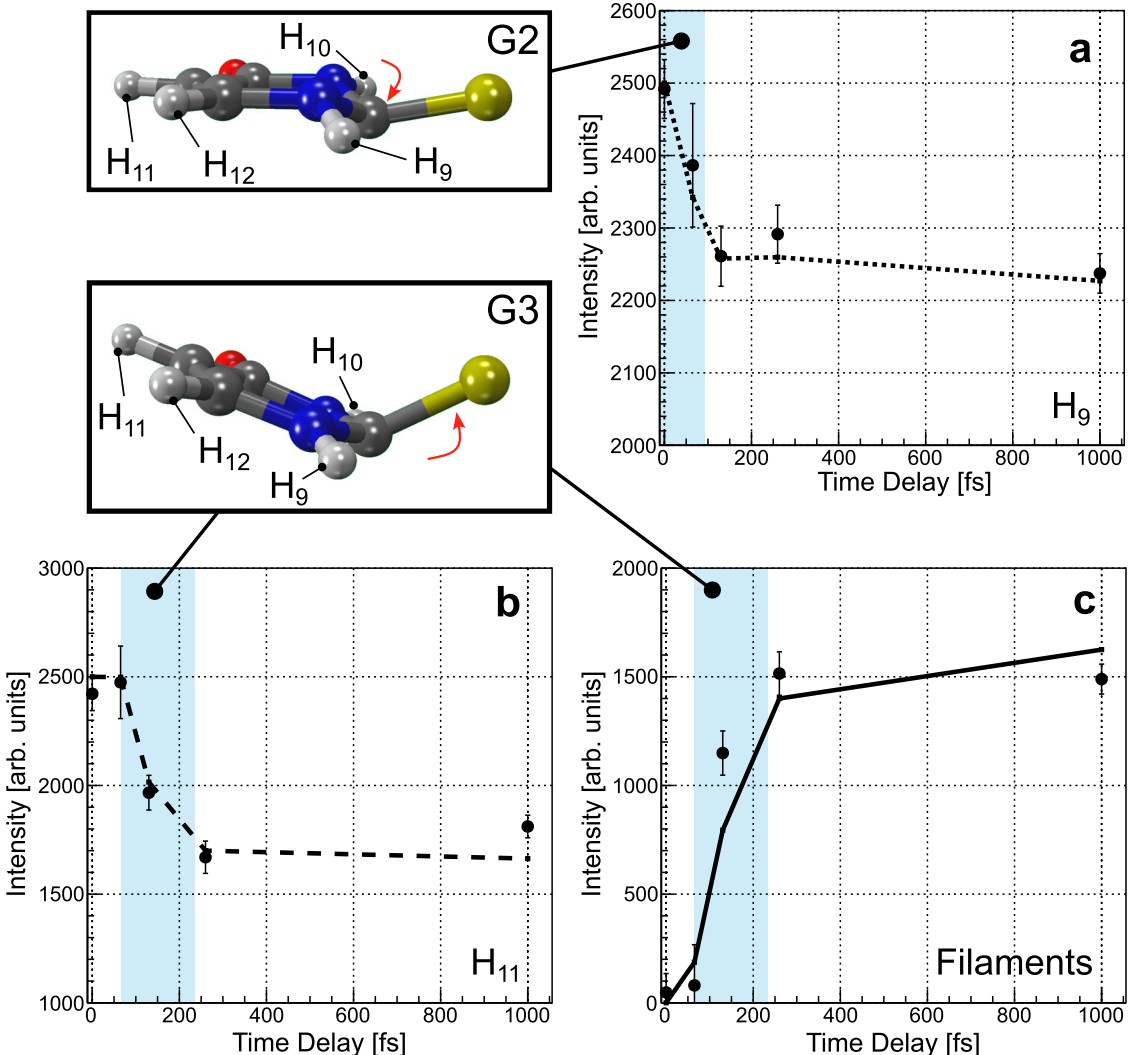

**Fig. 5 | Integrated intensity extracted from Fig. 4, where the integration regions are marked by boxes of different line styles. a, b** Angular emission regions of the $H_9$ and $H_{11}$ ions as a function of the pump-probe delay. **c** Corresponding plot for the "filament region". The instant change of intensity in panel (**a**) corresponds to a change in the molecular geometry G2 (top left) and the delayed onset in panels (**b**, **c**) to the one depicted as G3 (middle left). The dots show the experimental results (with error bars showing the standard deviation of the statistical errors), the lines correspond to the outcome of our simulations. Note: the experimental results have been scaled, see Methods section for details. The blue shaded areas highlight the instantaneous (**a**) and delayed (**b**, **c**) change of the intensity.

reveal striking details important for the in-depths understanding of internal conversion on the UV-induced dynamics of a molecule as large as twelve atoms.

## Methods

The experiment was performed at the SQS scientific instrument of the European X-ray Free-Electron Laser using cold target recoil ion momentum spectroscopy (COLTRIMS)[23,24]. A state-of-the-art COL-TRIMS reaction microscope is available at SQS as a permanent user endstation. In the following, we describe the experimental setup, the properties of the UV laser used for the excitation of the molecules and the X-rays employed for triggering the Coulomb explosion as well as details on the data analysis and the Coulomb explosion simulation. Details on the modeling of the molecular dynamics, which yielded the trajectories used as an input to our Coulomb explosion simulation can be found elsewhere[18,19].

### Experimental setup

We used the COLTRIMS technique to measure the momenta of several charged atomic fragments, which were generated in the Coulomb explosion of 2-thiouracil molecules, in coincidence. At room temperature, the target molecules exist in the form of a powder. Accordingly, in order to generate a supersonic gas jet consisting of 2-thiouracil molecules, we evaporated the substance in an oven, which was part of a nozzle generating the gas jet. The oven was held at a constant temperature of 240 °C, without degenerating the molecular sample. The supersonic jet formed as a mixture of the vapor and He carrier gas (which was applied with a stagnation pressure of 0.5 bar to the oven) expanded through the nozzle hole (i.e., an aperture with a diameter of 200 μm) into vacuum. Before entering the main chamber housing the COLTRIMS analyzer, the gas jet was skimmed/collimated in four stages to form a well-localized target. The third collimation stage is typically used to reduce the amount of target such that the coincident measurement of ions originating from a single molecule can be performed. The target jet was crossed at right angle with the light from the XFEL and the exciting UV laser. Ions which were created in the target region upon the interaction with the ionizing light were guided by an electric spectrometer field to a time- and position-sensitive microchannel-plate detector with hexagonal delay-line position readout and an active diameter of 120 mm. The spectrometer consisted of an acceleration

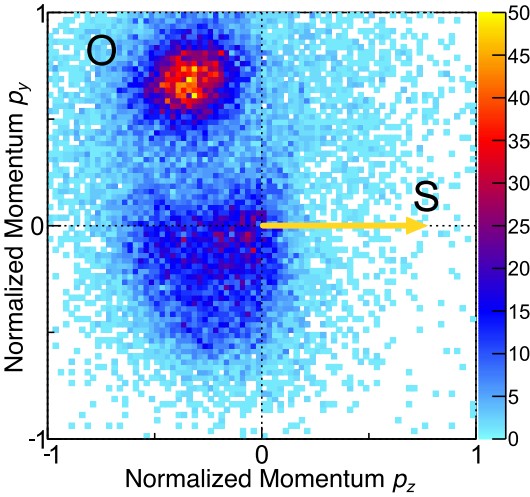
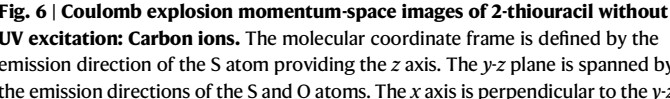
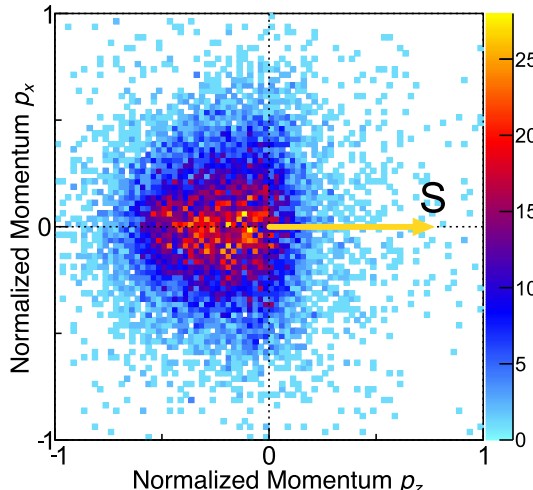

**Fig. 6 | Coulomb explosion momentum-space images of 2-thiouracil without UV excitation: Carbon ions.** The molecular coordinate frame is defined by the emission direction of the S atom providing the $z$ axis. The $y$-$z$ plane is spanned by the emission directions of the S and O atoms. The $x$ axis is perpendicular to the $y$-$z$ plane. Left: "Top view" projection of the measured momentum-space data of the oxygen atom and the carbon ions. Right: "Side view" projection of the measured carbon ion momenta. The momenta have been normalized such that the magnitude of the sulfur momentum equals 1 for each molecular ionization event.

region (covering the interaction volume of the gas jet and the light beams) with a length of 60 mm and a strong electric extraction field of 330 V/cm applied. The acceleration region was followed by a drift region (with a length of 120 mm), which was passed by the ions before they hit the detector. The ions detected with respect to each XFEL pulse were recorded in coincidence. We reconstructed the trajectory of each individual ion inside the spectrometer from its flight time and the position of impact in an offline analysis and deduced from this information the mass-over-charge ratio of the ion and its initial momentum vector after the Coulomb explosion.

### Properties of the UV laser and the X-rays

The European X-ray Free-Electron Laser provides trains of short X-ray pulses at a repetition rate of 10 Hz with train/burst duration of up to 600 μs. It was operated in a mode which yielded a maximum rate of pulses within such a train of 1.1 MHz. In order to incorporate the flight time of heavy molecular fragment ions, we employed every sixth of these pulses for our experiment yielding approximately 80 XFEL pulses per train in our measurement (and thus an effective repetition rate of 800 Hz). As a photon energy for the Coulomb explosion, we chose hν = 2.6 keV, well above the sulfur K-edge, which is situated at 2472 eV for atomic sulfur[48]. We obtained a single-shot pulse energy of approximately 2 mJ, which was measured by a gas monitor detector upstream of the beamline. The transmission of the beamline was estimated to be 85% for this range of photon energies and the given configuration[49], which yielded a pulse energy of approximately 1.7 mJ on target. The focus diameter of the X-ray beam at the interaction volume was tuned by looking at the high-charge yield from multiphoton ionization of Xe atoms inspecting the corresponding time-of-flight spectra and estimated as 5 μm FWHM based on an off-line microscopy characterization under similar conditions. The X-ray pulse length was as low as 8–10 fs determined indirectly by analyzing the spectral distribution of the SASE pulses[50].

The third harmonic of an optical laser operating at 800 nm and synchronized to the X-ray pulses was used for the excitation of the molecules[51]. Pulses with energies up to 8.5 μJ of 266 nm radiation were delivered to the interaction volume. The duration of the UV pulses was measured to be about 35 fs and the focus was set to a diameter of approximately 120 μm. The molecular absorption cross section is in the range of 30 Mbarn[52], leading to a saturation of the $S_0$-$S_2$ molecular transition in the center of the UV focus. We note that two-photon ionization by the UV beam does excite a part of the population in the

cationic ground state from the one-photon excited $S_2$ state. However, this does not interfere with the observation of deplanarization in the excited state, as the cationic ground state of 2-thiouracil is expected to remain planar. In this cationic state, the $\pi^*$ orbital localized on the $C_2$ atom that leads to deplanarization is not occupied[53]. The planar cationic ground state has been confirmed by an optimization at the MS(2)-CASPT2(11,9)/cc-pVDZ level of theory. The synchronization between FEL and laser pulses can be as good as 10 fs[54].

### Data analysis

As indicated above, molecular fragment ions originating from a single molecule were detected in coincidence and their vector momenta were obtained from their impact positions on the detector and their flight times after the photoreaction. The mass-to-charge ratio of the measured ions is determined from their measured flight times. Then the full momentum information is retrieved from the recorded data by reconstructing the ions' trajectories inside the COLTRIMS spectrometer for each individual ion. From the momenta, all derived quantities such as ion kinetic energies and emission angles are deduced. In particular, due to the coincident detection of the ions, relative emission angles between different ions are obtained. This advantage of a COLTRIMS measurement is used to define two different molecular coordinate frames, which are used to present the data. Firstly, the momentum of the $S^+$ ion acts as the $z$ axis of the coordinate frame, and the $y$-$z$ plane is spanned by taking in addition the momentum of the $O^+$ ion into account. By definition, the $y$ component of the $O^+$ momentum is greater than or equal to zero. The $x$ axis is perpendicular to the $y$-$z$ plane. The laboratory-frame momenta are then transformed into this molecular frame for each individual molecule which was exploded by the XFEL. In addition to this coordinate transformation, all momenta were scaled by the magnitude of the $S^+$ momentum prior to plotting. Secondly, in order to obtain the angular emission distributions shown in Figs. 2 and 4, a similar molecular coordinate frame is employed, which uses instead of the sulfur and oxygen ion momenta the sum and difference of these two momenta as a reference. Within this (much less intuitive) coordinate frame, the molecule is approximately oriented as indicated in Fig. 2 (for cases where it is in its planar ground-state equilibrium geometry).

The data presented in this article consist of threefold coincidences, i.e., a coincident detection of an $S^+$, an $O^+$, and a $H^+$ ion. In line with previous results obtained at the EuXFEL[28], we conclude that the rapid charge-up triggered by the XFEL occurs along a well-defined route

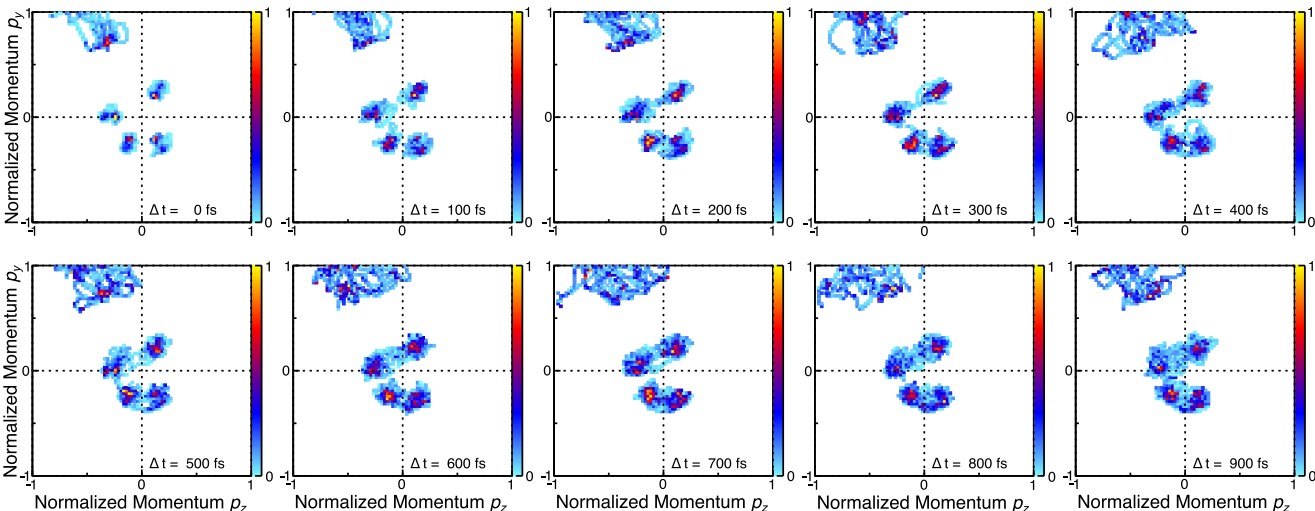

**Fig. 7 | Time-dependent Newton plots showing simulated data, *y-z* plane.** Results from our Coulomb explosion simulation for a sequence of time steps (indicated in each panel at the bottom, right) as Newton plots (i.e., in the same representation used for the measured results in Fig. 2a). The color bar depicts intensity in arbitrary units.

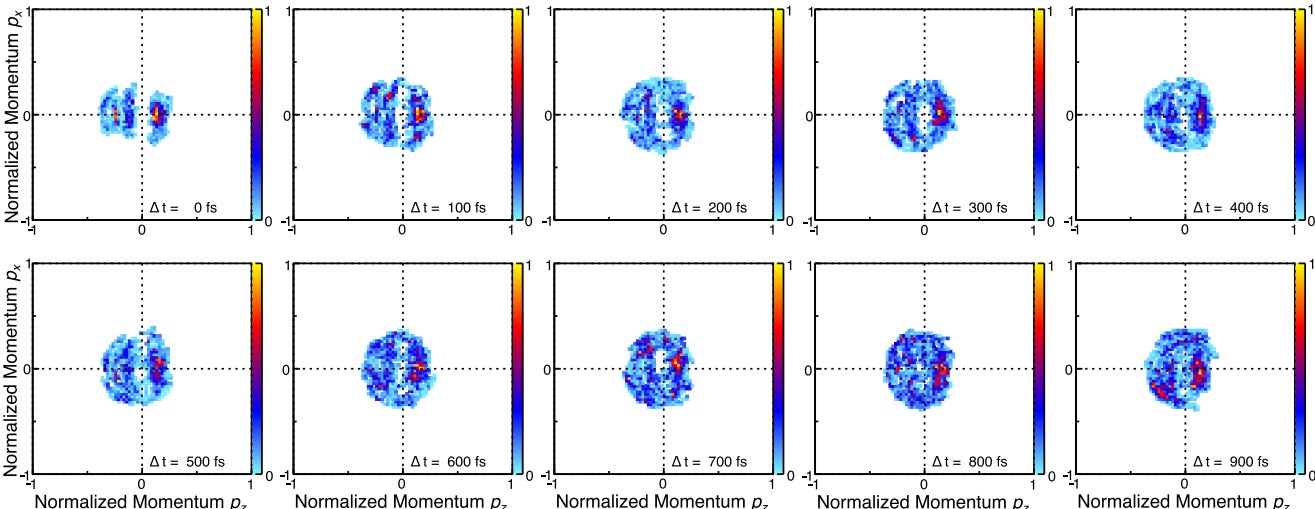

**Fig. 8 | Time-dependent Newton plots showing simulated data, *x-z* plane.** Results from our Coulomb explosion simulation for a sequence of time steps (indicated in each panel at the bottom, right) as Newton plots (i.e., in the same representation used for the measured results in Fig. 2b). The color bar depicts intensity in arbitrary units.

resulting in Coulomb explosion patterns which are (in particular) stable on a shot-to-shot basis. Therefore, a full picture of the Coulomb explosion is obtained even though only a subset of all generated ions is detected for each XFEL shot. Other than reported earlier[32], the charge-up observed in the present experiment was, however, less well-defined. We, therefore, rely in the main part of the article on the momentum-space Coulomb explosion patterns of the protons (inspected in a coordinate frame provided by the momenta of the O and S ions). The protons show the well-defined distributions visible in Fig. 2. The corresponding Newton diagrams showing the $C^+$ momenta in the same molecular coordinate frame are depicted in Fig. 6 and show only a rather broad and comparably featureless distribution. Our work, therefore, highlights the possibility to employ the protons (which are emitted very rapidly at the beginning of the charge-up by the XFEL) as snapshot-like messengers conveying details on the molecular geometry even in cases where the full charge-up does not yield a picture that is easy to interpret without employing sophisticated theoretical modeling.

In the experiment, we recorded separate datasets at pump-probe delays of 65 fs, 130 fs, 260 fs, and 1 ps and without UV pump (i.e., X-rays only). We recorded a similar amount of statistics for each delay step, with the exception of the dataset belonging to a pump-probe delay of

65 fs, which has less statistics (as indicated, e.g., by the corresponding statistical error bar in Fig. 5). In order to compare the different datasets, we normalized the measured data by the integrated number of shots recorded for each set. In Fig. 5, we integrated over certain regions of the angular emission distributions of the protons as obtained from our measurements and our Coulomb explosion simulation. The absolute values retrieved from this integration cannot be compared between the simulation and the experiment. In addition, other than in the simulation, there is a strong contribution from unpumped molecules in the measured distribution. Accordingly, only the shape of the time dependence of the integrated intensities can be compared between experiment and simulation, and the experimental data points have been scaled with respect to the simulation. The error bars provided in Fig. 5 correspond to the statistical error of each integrated value after being scaled accordingly.

### Simulated Coulomb explosion

We employed the trajectory data published earlier[19] as an input for a simple Coulomb explosion model in order to compare the resulting momentum-space information with our measured time-resolved Coulomb explosion results. The used trajectory data covered the

evolution of the molecule for a time of 1 ps after the UV excitation in time steps of 5 fs. Our Coulomb explosion code generates the momentum-space distribution of all atoms of the molecule by solving Newton's equations of motion for each time step of the provided trajectories after initializing each atom of the molecule at the given geometry with a charge of +1. Accordingly, our CE model assumes an instantaneous Coulomb explosion of the molecule into singly charged atomic fragments with purely Coulombic interactions. As the real Coulomb explosion is not instantaneous, a main difference between the modeled and the measured results is (typically) the overestimation of the kinetic energy of the ions after the Coulomb explosion. This discrepancy vanishes mostly when considering normalized momenta as depicted in Fig. 2. The angular emission pattern is affected by the model assumptions, as well, but (as the results presented in Fig. 4 indicate) to a lesser extent than the final-state energies. Figure 7 shows the temporal evolution of the proton and oxygen ion momenta as obtained from the Coulomb explosion simulation in the form of Newton plots (same representation as in Fig. 2a). Figure 8 shows the simulated proton momenta in correspondence to Fig. 2b.

## Data availability

Data recorded for the experiment at the European XFEL are available at https://doi.org/10.22003/XFEL.EU-DATA-003155-00. The data generated in this study are provided in Source Data files. Source data are provided with this paper.

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

## Acknowledgements

We acknowledge European XFEL in Schenefeld, Germany for the provision of XFEL beam time at the SQS instrument and would like to thank the staff for their assistance. HVSL, AR, and DR were supported by the Chemical Sciences, Geosciences, and Biosciences Division, Office of Basic Energy Sciences, Office of Science, US Department of Energy under grant no. DE-FG02-86ER13491. SB and KC were supported by grant no. DE-SC0020276 from the same funding agency, and AD and ASV by grant no. PHYS-1753324 from the National Science Foundation. FT acknowledges funding by the Deutsche Forschungsgemeinschaft (DFG, German Research Foundation) - Project 509471550, Emmy Noether Programme. HI acknowledges the Natural Sciences and Engineering Research Council of Canada and the NRC Quantum Sensors Project. M.M. acknowledges support by the Cluster of Excellence 'Advanced Imaging of Matter' of the DFG—EXC 2056 and project ID 390715994. AEG acknowledges funding by the European Union under project 101067645. SM acknowledges funding from the Austrian Science Fund (FWF), grant https://doi.org/10.55776/P25827. MG acknowledges DFG funding via Grant GU 1478/1.

## Author contributions

T.J., S.B., K.C., R.B., M.E.C., S.D., U.F., A.E.G., M.I., R.I., G.K., H.V.S.L., F.L., T.Ma., T.Mu., Y.O., B.S., A.-T.-N., S.U., A.S.V., A.R., D.R., M.M., H.I., and M.G. performed the experiment. T.J. analyzed the experimental data and the Coulomb explosion simulation results. S.M. simulated molecular trajectories. S.B., K.C., H.V.S.L., A.S.V., A.R., and D.R. provided the Coulomb explosion code and performed the Coulomb explosion of the simulated molecular trajectories. T.J., H.I., and M.G. discussed the data and simulations and worked on an interpretation. T.J., S.M., H.I., and M.G. produced the figures and wrote the first draft of the paper. All authors (including F.T. and D.M.) contributed to the final version of the paper in iterative discussions.

## Funding

## Competing interests

The authors declare no competing interests.

## Additional information

[1]Max-Planck-Institut für Kernphysik, Heidelberg, Germany. [2]European XFEL, Schenefeld, Germany. [3]Institute of Theoretical Chemistry, Faculty of Chemistry, University of Vienna, Vienna, Austria. [4]James R. Macdonald Laboratory, Kansas State University, Manhattan, Kansas, USA. [5]Clinical Pharmacology Unit, IRCCS Azienda Ospedaliero-Universitaria di Bologna, Bologna, Italy. [6]Deutsches Elektronen-Synchrotron DESY, Hamburg, Germany. [7]Stanford PULSE Institute, SLAC National Accelerator Laboratory, Menlo Park, California, USA. [8]Department of Physics, Universität Hamburg, Hamburg, Germany. [9]Department of Chemistry, University College London, London, UK. [10]Goethe-Universität Frankfurt, Frankfurt am Main, Germany. [11]Fritz-Haber-Institut der Max-Planck-Gesellschaft, Berlin, Germany. [12]Advanced Laser Light Source @ INRS, Centre Énergie, Matériaux et Télécommunications, Québec, Canada. [13]Department of Physics, University of Ottawa, Ottawa, ON, Canada. [14]Institute of Physical Chemistry, Universität Hamburg, Hamburg, Germany. ✉e-mail: till.jahnke@xfel.eu; markus.guehr@desy.de

