## [Transparent Peer Review file · Nature Communications]

Direct observation of ultrafast symmetry reduction during internal conversion of 2-thiouracil using Coulomb explosion imaging

Corresponding Author: Professor Till Jahnke

Version 0:

Reviewer comments:

Reviewer #1

(Remarks to the Author)

The authors report on time-resolved Coulomb explosion imaging, induced by a fs x-ray pulse, of intramolecular dynamics of 2-thiouracil following 1-photon excitation by an ultraviolet fs laser pulse. By comparing the experimental results with simulations of the Coulomb explosion, employing previously published trajectory calculations as input, the authors obtain novel and direct insight into the deplanarization of the molecules after the UV-induced electronic excitation. Experimentally, it is shown how the emission directions of the H⁺ ions are particularly useful for characterizing the structural change of the molecule.

In my opinion, this is an interesting piece of work that demonstrates the rich information that timed Coulomb explosion imaging is capable of providing – here about an important photoinduced intramolecular transformation, which is of relevance and importance to processes in biomolecules and to photocatalysis. The results are convincing and point towards future exciting opportunities, the manuscript is well-written and it should be of interest to a broad readership. As such, the manuscript is suitable for Nature Communication and I recommend its publication when the authors have addressed the following comments.

Page 1, last paragraph

'Heterocyclic molecules generally absorb light via a strong $\pi \rightarrow \pi^*$ transition as shown in Fig. 1A.' I think it is fine to label the excited state ' $\pi \rightarrow \pi^*$ ' but the transition should be called ' $\pi \rightarrow \pi^*$ ' / ' $\pi \rightarrow \pi^*$ '

Page 4, first paragraph

'The inversion of the measured momentum-space data to real space is viable for small molecules [24-27].'

I suggest that one or both of the following very recent papers are cited to reflect the breadth of Coulomb explosion imaging to retrieve bond distances and internuclear wavefunctions for small molecules- here of di- and triatomic systems that were not addressed before.

H. H. Kristensen et al. PHYSICAL REVIEW A 107, 023104 (2023)

DOI: 10.1103/PhysRevA.107.023104

L. Kranabetter et al. J. Chem. Phys. 160, 131101 (2024)

DOI: <https://doi.org/10.1063/5.0200389>

Page 5, Fig. 2C

I suggest you label all atoms with numbers (as in Fig. 1) on the molecular sketch since you refer to them later in the manuscript. You could also consider to include the numbers on the molecular sketches in panels A and B.

Page 6, Fig. 3A: This is quite a poor figure. It is very difficult to see what is going on. Please improve it significantly.

Page 6, Fig. 3B

Why are you using $\cos\theta$ rather than just θ ? I think many readers would find it more intuitive to see θ used as the variable – notably when trying to see what this means for the molecular structure with reference to the definition of θ and ϕ in Fig. 2C. I suggest to change to θ and update the text accordingly.

Page 6, last paragraph

'Simulations agree with the UV-induced change of the relative O-S angle observed in the experiment.' This feels a bit of an early statement since the experimental results have not yet been presented

Page 6, last paragraph

'for later delays' -> 'for larger delays' (or 'for later times')

Bottom of page 6 and top of Page 7

The agreement between the simulated and the experimental results in Fig. 3C seems quite good. However, an explanation about what can be concluded about the change of the molecular structure (the S-O angle), based on the observed normalized intensity difference, is missing. Please add some sentences to clarify what you can conclude from Fig. 3C. (For instance, $\cos\alpha = -0.4$ corresponds to 114 deg, which seems to be the initial angle between the C-O and C-S bonds)

Page 7, Fig. 4

It is unfortunate to use red and green for the lines and squares. Colorblind people cannot tell the difference. Please use a different coloring.

For the colorbar scale at the right: Please add numbers as in Fig. 2C. This should, by the way, also be changed on Fig. 3A.

Page 8, line -6

'starts with a slight delay, as' -> 'starts after a slight delay, as'

Page 8, Fig. 5

Add label-numbers on panel G2 and G3 if possible.

Page 9, line 2: 'At later delays' -> 'At larger delays'

Page 9, Conclusion, first line

Add 'according to theory' after 'which'

Reviewer #2

(Remarks to the Author)

This study presents an experimental scheme based on X-ray Coulomb explosion imaging, aimed at monitoring out-of-plane distortions in heterocyclic compounds following electronic excitation. This technique holds the potential to significantly contribute to the interpretation of UV-induced dynamics. Although the title and abstract successfully capture the reader's attention, I found the manuscript lacks clarity regarding how this method aids in observing out-of-plane distortions during excited-state non-adiabatic processes, particularly for non-specialists. Below, I offer a more detailed explanation of my concerns, along with several minor suggestions for improvement.

If protons and other atoms, including O and S in this specific case, can (significantly) oscillate out of the molecular plane at any stage of the molecular dynamics, leading to a loss of the Cs original Franck-Condon symmetry, then, how can this method infer the out-of-plane deviations of specific second-row or heavier atoms based on proton emission directions, at specific critical points in the deactivation process? In other words, how does vibrational excitation (even more when the sample is heated in the set up) affects the emission direction of protons?

In line with this, would it be feasible to estimate sulfur dihedral angle relative to the molecular plane for particular pump probe delays? In cases like 2-thiouracil, where multiple competing pathways are involved, how sensitive is this technique to branching yields?

For the specific system discussed here, ref [17] indicates that at the ADC(2) level of theory—used for the semiclassical simulations—sulfur diverts significantly out of plane at various points:(i) at the S_2 minimum, (ii) at the S_2/S_1 intersection, (iii) at the intersystem crossing (ISC) point, and (iv) at the first pp^* triplet minima reached.

Given this information, how can this approach provide information on the internal conversion process? Additionally, how sensitive is this method to variations in the dihedral angle?

The authors also suggest that this technique could be applied to the monitoring of pp*/ps* crossings. However, the pp*/ps* crossings that come to mind are typically planar. Would this then involve using the alpha angle instead? A brief discussion on this point would be appreciated.

Minor Comments:

1. References 6 and 7 in the first paragraph seem overly specific to nucleic acids, which feels slightly out of tune with the broader focus of the introduction. I recommend either replacing these with more general references or incorporating additional citations for balance.
2. Figure 2D is complex and not self-explanatory. I suggest providing a brief explanation in the text to aid readers. Additionally, the term "binding energy" appears several times and would benefit from some contextualization.
3. Including a graphical representation of the alpha angle in one of the schemes presented would be helpful for readers.

Reviewer #3

(Remarks to the Author)

The manuscript has been professionally, clearly and carefully written and the authors obviously mastered their experimental technique. They also do a good job explaining the background of chemical reactivity, internal conversion and its link to symmetry breaking by nuclear motion. This scientific theme is well established here and shows that the paper is of high contemporary interest and that the out-of-plane distortions are an important aspect of the relaxation mechanisms of a large class of molecules. The analysis of the data has been performed carefully and the quality of the results, in terms of CEI, are very good. The conclusions from the experimental data are well-formulated and show that the paper clearly contributes to our picture of photoinduced dynamics in small quantum systems.

On the other hand, the authors also stress repeatedly that they are using a new experimental method that opens new avenues for research (e.g. "Our experimental scheme opens new avenues for time-resolved studies of complex molecules in the gas phase"). This presumably enhances the novelty factor of their submission. In contrast to the convincingness of their scientific claims, the instrumental/methodological ones are not very convincing. The Coulomb Imaging technique (CEI), while relatively new, has been developed actively during the past years (involving the present authors) and there are several publications devoted to this technique as a means to obtain (also time-resolved) "snapshots" of the structural information of a single molecule. This is by now well-known in the community, and CEI is hardly a novel technique anymore. Also, as supposedly novel details, it is for example already known that the hydrogen ions can be an excellent probe in the CEI of the molecular geometry and that CEI can be successfully coupled with a pump-probe scheme (just as one example, see Crane, S. W., Lee, J. W. L., Ashfold, M. N. R. & Rolles, D. Molecular photodissociation dynamics revealed by Coulomb explosion imaging. *Phys. Chem. Chem. Phys.* 25, 16672–16698 (2023).) Other than that, it is not clear, what exactly is the outstanding experimental and methodological novelty of this study, apart from the normal evolution of a technique. Of course, the system studied is new, and perhaps more complex than the systems before (as a natural evolution of the method). Also, the particular geometry change -- the planar symmetry breaking -- has perhaps not been observed before by CEI. But it remains unclear, how this constitutes a qualitative important new step, compared to observing any other geometry change. From statements like "While we demonstrate the validity of the time-resolved X-ray CEI approach to molecular symmetry ...", it is not clear, why would molecular symmetry change be such a special circumstance from the viewpoint of CEI?

In this referee's opinion, the authors have overstated their instrumental/methodology novelty claims and should either be more specific about why this study constitutes a very important advancement over what is already established, or tone down these claims. Whether the scientific results warrant publication in *Nature* or in a more specialized high-impact journal, has no obvious answer.

Specific comments:

p. 3: "The discussed internal conversion requires an out-of-plane molecular motion belonging to the A' irreducible representation." This was not immediately clear to this reviewer. Is it a specific vibrational normal mode of A' or is this connection explained on some other way? This statement would benefit from clarification/illustration.

Version 1:

Reviewer comments:

Reviewer #1

(Remarks to the Author)

The authors have addressed my comments and criticism in a satisfactory manner. I deem the manuscript suitable for publication in *Nat. Comm.*

Henrik Stapelfeldt

Reviewer #2

(Remarks to the Author)

After reviewing the authors' responses to the referees' comments and evaluating the revisions made to the manuscript, this

reviewer believes the work is now suitable for publication.

Reviewer #3

(Remarks to the Author)

The authors have responded adequately to the points raised. Indeed, their response confirms the impression that technique-wise the results represent rather a normal evolution of CEI towards more complex systems and various aspects of molecular geometry (such as symmetry properties). With the modified "novelty claims" and taking into account the high general relevance of the study, I have no objections to publishing the manuscript in its present form.

Dear editor,

We would like to thank the referees for reviewing our manuscript and providing their assessment and very constructive criticism. The reports helped us to improve our paper substantially. Below you find point by point answers to the issues raised by the three referees. Our answers are given in blue, (larger) changes to the manuscript are given in green. We have generated final vector-graphics versions of our figures and amended them as suggested by the referees.

We hope that with the changes made, the paper is now suitable for publication in Nature Communications.

Markus Gühr

Reviewer #1 (Remarks to the Author):

The authors report on time-resolved Coulomb explosion imaging, induced by a fs x-ray pulse, of intramolecular dynamics of 2-thiouracil following 1-photon excitation by an ultraviolet fs laser pulse. By comparing the experimental results with simulations of the Coulomb explosion, employing previously published trajectory calculations as input, the authors obtain novel and direct insight into the deplanerization of the molecules after the UV-induced electronic excitation. Experimentally, it is shown how the emission directions of the H⁺ ions are particularly useful for characterizing the structural change of the molecule.

In my opinion, this is an interesting piece of work that demonstrates the rich information that timed Coulomb explosion imaging is capable of providing – here about an important photoinduced intramolecular transformation, which is of relevance and importance to processes in biomolecules and to photocatalysis. The results are convincing and point towards future exciting opportunities, the manuscript is well-written and it should be of interest to a broad readership. As such, the manuscript is suitable for Nature Communication and I recommend its publication when the authors have addressed the following comments.

We would like to thank the referee for the support of our paper and the suggestions, which we address in detail below.

Page 1, last paragraph

'Heterocyclic molecules generally absorb light via a strong $^1\pi\pi^*$ transition as shown in Fig. 1A.' I think it is fine to label the excited state ' $^1\pi\pi^*$ ' but the transition should be called ' π to π^* ' / ' $\pi \rightarrow \pi^*$ '

We have changed the corresponding sentence accordingly.

Page 4, first paragraph

'The inversion of the measured momentum-space data to real space is viable for small molecules [24-27].' I suggest that one or both of the following very recent papers are cited to reflect the breadth of Coulomb explosion imaging to retrieve bond distances and internuclear wavefunctions for small molecules- here of di-and triatomic systems that were not addressed before.

H. H. Kristensen et al. PHYSICAL REVIEW A 107, 023104 (2023) DOI: 10.1103/PhysRevA.107.023104

L. Kranabetter et al. J. Chem. Phys. 160, 131101 (2024) DOI: <https://doi.org/10.1063/5.0200389>

We have added the two references to our paper.

Page 5, Fig. 2C

I suggest you label all atoms with numbers (as in Fig. 1) on the molecular sketch since you refer to them later in the manuscript. You could also consider to include the numbers on the molecular sketches in panels A and B.

We have changed the figure in line with the suggestion by Referee #1. The full set of numbers is now included in the sketch of the molecule in Panel A, and we labeled the protons in addition in Panel C. Together with these changes, we replaced the sketch of the spherical coordinate frame with a rendered version, which is hopefully easier accessible. We have added an additional axis to the right panel of Fig. 2C showing the angle Θ in degrees. Please see the below answer (concerning "Page 6, Fig.3B") for more details.

Page 6, Fig. 3A: This is quite a poor figure. It is very difficult to see what is going on. Please improve it significantly.

We were a little unsure what exactly to change given the comment by the referee. We therefore performed a rebinning of and a smoothing of the trajectory data. This makes it indeed easier to observe the overall time-dependent trends of $\cos(\alpha)$. We added in addition a small icon indicating the definition of angle α . We added a corresponding description to the manuscript:

The Coulomb explosion results have been smoothed in order to highlight the overall trends despite the discrete nature of the modeled trajectory dataset.

Page 6, Fig. 3B

Why are you using $\cos\theta$ rather than just θ ? I think many readers would find it more intuitive to see θ used as the variable – notably when trying to see what this means for the molecular structure with reference to the definition of θ and ϕ in Fig. 2C. I suggest to change to θ and update the text accordingly.

We employed $\cos(\alpha)$ (and later in the manuscript $\cos(\Theta)$ for the representation in spherical coordinates) as this incorporates correctly the solid angle-element. In a pure representation in α (or Θ), events lying close to to poles (i.e. close to 0° and 180°) are largely underestimated. Another option to avoid this problem would be to weight the histogram data by $1/\sin(\Theta)$. This, however, typically leads to “exploding” error bars close to the poles. We therefore decided to stick to the cosine-representation - even though it is (as pointed out by the referee) a less intuitive representation when trying to understand the geometrical implications of the histograms in detail. However, to aid the readers here, we added an additional “distorted” axis to Figures 2C(right), and 3A-C showing the angle in degrees, as well. We hope that this compromise is in line with the referee’s suggestion.

Page 6, last paragraph

‘Simulations agree with the UV-induced change of the relative O-S angle observed in the experiment.’ This feels a bit of an early statement since the experimental results have not yet been presented.

We agree and removed this sentence, accordingly.

Page 6, last paragraph

‘for later delays’ -> ‘for larger delays’ (or ‘for later times’)

We replaced “later” with “larger” as suggested.

Bottom of page 6 and top of Page 7

The agreement between the simulated and the experimental results in Fig. 3C seems quite good. However, an explanation about what can be concluded about the change of the molecular structure (the S-O angle), based on the observed normalized intensity difference, is missing. Please add some sentences to clarify what you can conclude from Fig. 3C. (For instance, $\cos\alpha = -0.4$ corresponds to 114° , which seems to be the initial angle between the C-O and C-S bonds).

This is a very interesting observation made by the referee. We are frankly quite surprised to see such a good agreement between the position-space angle between the C-O and C-S bonds and the measured relative emission angle of the S^+ and the O^+ . In particular, work on smaller molecules showed in the past that the measured emission direction can be largely skewed due to several effects. For example, the long-range character of the Coulomb repulsion of the fragments plays a role, and the “non-pointlike” character of the introduced charge needs to be considered as well. The latter is, for example, addressed sometimes by employing fractional charges in simulations. We added a corresponding sentence to the manuscript and provide a reference to pioneering work on this topic (Saito et al., JESRP 141, 183–193 (2004)):

Interestingly, the peak of the measured angle α is very close to the initial angle between the C-O and C-S bonds, which is 116° in the S_0 state [17]. Such good mapping from the initial position-space angle to momentum space by the Coulomb explosion is surprising, as already from smaller molecules it is known that there are several properties of a *real* Coulomb explosion (long-range repulsion, charge distribution is not point-like), which typically skew the finally observed emission angles [Saito2004]. Finally, the angle between the C-O and C-S bonds approaches 104° . This is in line with the increase in intensity in the range of $\cos(\alpha)=0.1$ for later times in Fig. 3C

Page 7, Fig. 4

It is unfortunate to use red and green for the lines and squares. Colorblind people cannot tell the difference. Please use a different coloring. For the colorbar scale at the right: Please add numbers as in Fig. 2C. This should, by the way, also be changed on Fig. 3A.

We have changed Figure 4 and replaced the color-labeling by using full, dotted and dashed line-boxes instead. We amended the text accordingly. As the data plotted in the panels is normalized, the absolute values are arbitrary. We therefore added a range from 0..1 to the color scale of Fig. 4 and labeled the scale as "Normalized Intensity [arb. units]" as in Figure 3. In Fig. 3 we added values "0..1" to the color bar, as well.

Page 8, line -6

'starts with a slight delay, as' -> 'starts after a slight delay, as'

We have amended this sentence as requested.

Page 8, Fig. 5

Add label-numbers on panel G2 and G3 if possible.

We have added labels for the protons to G2 and G3 and changed the line styles of the (formerly colored) lines indicating the results from theory to comply with those from Fig. 4.

Page 9, line 2: 'At later delays' -> 'At larger delays'

We have changed "later delays" to "larger delays".

Page 9, Conclusion, first line

Add 'according to theory' after 'which'

We added "according to theory" as requested.

Reviewer #2 (Remarks to the Author):

This study presents an experimental scheme based on X-ray Coulomb explosion imaging, aimed at monitoring out-of-plane distortions in heterocyclic compounds following electronic excitation. This technique holds the potential to significantly contribute to the interpretation of UV-induced dynamics. Although the title and abstract successfully capture the reader's attention, I found the manuscript lacks clarity regarding how this method aids in observing out-of-plane distortions during excited-state non-adiabatic processes, particularly for non-specialists. Below, I offer a more detailed explanation of my concerns, along with several minor suggestions for improvement.

We thank the referee for acknowledging the power of the method. We are grateful for the critical remarks that we address below.

If protons and other atoms, including O and S in this specific case, can (significantly) oscillate out of the molecular plane at any stage of the molecular dynamics, leading to a loss of the Cs original Franck-Condon symmetry, then, how can this method infer the out-of-plane deviations of specific second-row or heavier atoms based on proton emission directions, at specific critical points in the deactivation process? In other words, how does vibrational excitation (even more when the sample is heated in the set up) affects the emission direction of protons?

The referee poses two questions here. 1) How is it possible to observe time-dependent symmetry distortions at all, as vibrational dynamics are present which intrinsically blur out the molecular structure. 2) How can the protons be sensitive to symmetry distortions of the second row elements?

Concerning the first question, Coulomb explosion imaging does in this respect not differ from other (ensemble averaging) imaging methods as, e.g., X-ray scattering. As indicated correctly by the referee, even in their ground states molecules show non-negligible fluctuations yielding position-space distributions, which are rather "thermal ellipsoids" than sharply peaked. Nonetheless, the thermal ellipsoid contains information about the molecular distribution in the ground state. From such thermal ellipsoids one can still estimate whether a molecule is, e.g., *on average* planar or not.

However, an additional aspect of CEI is that we do not retrieve position-space directly but the momentum distributions of the fragments after the explosion. While the inversion of the measured momentum space back to position space is (as pointed out in the manuscript) far from trivial, in many cases a zeroth-order assumption is that the fragments are emitted along their initial bonds (so-called axial recoil approximation). For example, Referee 1 pointed out (see above) that the measured relative emission angle between the O⁺ and S⁺ fragment is close to the initial angle between the C-O and C-S bonds. As our study indicates, this approximate behavior seems to hold, as well, for the protons (see Fig. 2A). Comparing the width of the proton angular emission distributions (e.g. looking at the H9 peak in Fig. 4, top

row) in the unpumped and the excited case shows directly that a difference between the ground state and the symmetry-broken geometry is observable in the Coulomb explosion momenta. We have visualized this once more by projecting out the distribution of angle ϕ in Fig. 4 for the case of $-0.8 < \cos(\theta) < -0.2$ and comparing the two extreme cases (unpumped and UV-excited, 1ps). The distributions are shown below (their integral has been normalized to 1 for comparison), the UV excited case has a much broader H9-peak (located at $\phi=0^\circ$) than the ground state (which correctly includes the vibrational GS-fluctuations). The “perfect” Franck-Condon geometry (i.e. without any vibrations) would yield a single spike at $\phi=0^\circ$.

In addition, we can see the effect of the “out-of-plane”-broadening of the proton momenta directly in the experiment, as well. The figure below shows the proton momenta in the molecular frame (as in Fig. 2B, i.e., the “side view” on the molecule) for the unpumped case (left) and the UV-excited case (right, longest pump-probe delay). It is clearly visible that the p_x momenta are more spread/less peaked in the latter case.

With respect to the second part of the question, the answer is twofold. First, if the sulfur moves out of the plane, any atom becomes a reporter since the molecular coordinate frame employed in our observations is spanned by the sulfur and oxygen sum- and difference momenta. Second, as the symmetry of the ring changes, the bond directions with the hydrogen changes (as we show in the geometry-images in Figs. 1C and 5,G2,G3), as well. The molecular symmetry change does (thus) not only correspond to a symmetry change involving the

carbons but also to one involving the hydrogens. This is the intuitive essence of employing the emitted protons as messengers for molecular geometry. Strictly speaking however, we cannot directly infer from the experimentally observed hydrogen symmetry change, that the carbon also changed its position. That is only possible by saying the sp hybridized orbitals also require the carbon next to the N-H bond to change position if the hydrogen bond changes its direction. We believe that this is not a very controversial assumption and in addition the comparison with the molecular dynamics simulations seems to verify this effect.

We added the following sentences to the introduction of the “Time-resolved results and simulations” section in order to further clarify our approach::

The reasoning behind this approach is twofold. First, if the sulfur moves out of the plane, any atom becomes a reporter since the molecular coordinate frame employed in our observations is spanned by the sulfur and oxygen sum- and difference momenta. Second, as the symmetry of the ring changes, the bond directions with the hydrogen changes, as well, as depicted by the geometry-sketches in Fig. 1C.

In line with this, would it be feasible to estimate sulfur dihedral angle relative to the molecular plane for particular pump probe delays? In cases like 2-thiouracil, where multiple competing pathways are involved, how sensitive is this technique to branching yields?

A direct extraction of the sulfur dihedral angle relative to the molecular plane is at the current level of maturity of the CEI technique not possible. As described in our manuscript the necessary inversion of the momentum-space data to position space is only doable for very small systems consisting of ~3 atoms. Even when assuming that the sulfur ion is always emitted along its bond, it is not straightforward to extract this angle from the momentum-space data, as a suitable coordinate frame (constructed from momenta of other explosion fragments) needs to be found, which represents “the molecular plane”. We provide an example of a simulation (which allowed us to pick such a distinct coordinate frame consisting of two of the hydrogens) below, which gives an impression on the conceptual sensitivity of CEI.

In detail, we performed a CE simulation of the molecular geometries along the minimum energy path depicted in Fig. 3 of the supporting information document of Mai *et al.*, *J. Phys. Chem. Lett.* **7**, 1978–1983 (2016). This figure is shown on the left. (Data used from ‘Intersystem Crossing Pathways in the Noncanonical Nucleobase 2-Thiouracil: A Time-Dependent Picture.’, by Mai, S., Marquetand, P., and González L., ref [18], DOI: 10.1021/acs.jpcllett.6b00616, copyright by American Chemical Society, disclaimer of warranties at https://pubs.acs.org/page/policy/authorchoice_ccby_termsfuse.html, licensed under CC-BY <http://creativecommons.org/licenses/by/4.0/>.) The simulated geometries correspond to the path from the nearly planar S₂/S₁ conical intersection to the pyramidalized S₁ minimum. The simulation consists (as in our manuscript) of an idealized, instantaneous Coulomb explosion into singly charged atomic fragments. It demonstrates that CEI is indeed a very sensitive technique. In the chosen molecular frame (here the protons H₁₁ and H₁₂ define the coordinate frame, the two molecular icons in the panels indicate the corresponding molecular orientation), the movement of the sulfur out of the plane is strongly visible (quarter-circle like feature in the right panel). This suggests that even small deviations should be traceable despite the momenta of the protons (which define the coordinate frame) are spread due to ground state vibrations+UV excitation, as well. In addition, further details are visible as, for example, the sulfur bond length changes along this path, which leads to a decreased momentum magnitude of the sulfur ion for the quarter-circle feature, indicating the rich information available in such momentum-space data. As the CEI is developing rapidly due to progress in FEL intensity, pulse lengths and also repetition rate, we expect that future experiments will be able to capture even more of these simulated details.

The CEI technique is still developing in large steps. Firstly, the real charge-up in an experiment is less violent than in the simulation, in particular for table-top laser-CEI. However, recent XFEL-based work reached new levels in this respect, but there are still many technological obstacles to take in order to approach the idealized case of our simulation. Even in the case of a full CE into atomic ions, it is not possible nowadays to detect all these ions in coincidence in a real experiment. This restricts, as well, the analysis of the measured data. For example, the coordinate frame chosen for the simulation above is at the current level of experiment not yet an option and alternative concepts need(ed) to be developed (as demonstrated in our paper). Nevertheless, we can make the most important statement about deplanarization, which is directly linked to the internal conversion.

With respect to branching yields, CEI is in many cases actually a favorable technique. It was in the past often suited for identifying minority species/pathways as these had distinct fingerprints in the recorded multi-dimensional momentum space. (As e.g. in our current experiment: we can infer information on the behavior of the sulfur by inspecting the correlated momenta of H⁺, O⁺ and S⁺ fragments, i.e. a nine-dimensional momentum space.)

For the specific system discussed here, ref [17] indicates that at the ADC(2) level of theory—used for the semiclassical simulations—sulfur diverts significantly out of plane at various points:(i) at the S₂ minimum, (ii) at the S₂/S₁ intersection, (iii) at the intersystem crossing (ISC) point, and (iv) at the first pp* triplet minima reached. Given this information, how can this approach provide information on the internal conversion process? Additionally, how sensitive is this method to variations in the dihedral angle?

The approach we follow in this work is a different one. As indicated above, it is not straightforward to obtain position-space information directly from the measured momentum space of the molecular fragments. In the present work we extract information on the molecular geometry from the combined momentum distributions of sulfur ions, oxygen ions and protons (which we measured in coincidence). This provides overall more information than a hypothetical measurement of just the dihedral angle of the sulfur (which is, as indicated by the referee, actually only a crude observable for the UV-excited molecular dynamics). With the combined momentum-space information, we are able to track essential parts of the internal conversion-route by observing fingerprints of the overall deformation of the molecule along that pathway. By tracking the molecular-frame angular distribution of the H9 ion, we can trace back the geometry change prior to the internal conversion (which consists only of small changes in the sulfur-angle). The H11 ion molecular-frame angular distribution changes after the internal conversion, which can be seen in our data, as well. That being said, an internal conversion is by definition an electronic process. Accordingly, we cannot trace it “directly” without measuring (e.g.) electrons. But we can trace the underlying geometry changes and relate them via the pump-probe delay to previous measurements, which targeted (solely) this electronic component in order to get a full (and consistent) picture. This is the approach we have followed in the present manuscript. In order to clarify this some more, we added the following:

While CEI itself does not provide direct information about the involved electronic states, we discover that the hydrogen atoms of the molecule are ideal messengers for the molecular symmetry.

The authors also suggest that this technique could be applied to the monitoring of pp^*/ps^* crossings. However, the pp^*/ps^* crossings that come to mind are typically planar. Would this then involve using the alpha angle instead? A brief discussion on this point would be appreciated.

We appreciate the question of the reviewer. We would like to point out that the same symmetry arguments that apply for the $pipi^*/npi^*$ internal conversion also apply for the $pipi^*/pisigma^*$ internal conversion. The strength of the needed symmetry breaking might be so weak, that indeed the ground state vibrations might be sufficient to reach this, and we guess that this is meant by the referee. With the current state of the art in the experiment, we can observe the symmetry breaking in the 2-tUra. We have not systematically studied the ultimate limits of the CEI sensitivity and this is beyond the scope of the current paper. However, to increase sensitivity, one can implement adiabatic cooling of the target via high-pressure expansion with a carrier gas, which has not been implemented in our case. This will reduce the ground state excitation, especially for low energy vibrational modes, and prepare a cleaner background for the optically induced deplanarization. In addition, as mentioned above, a part of today's effort when applying CEI to a problem lies in the identification of a suitable observable within the measured multi-dimensional momentum space. It is therefore hard to know beforehand which observable is most suited to target the symmetry-breaking of a pp^*/ps^* crossing.

Minor Comments:

1. References 6 and 7 in the first paragraph seem overly specific to nucleic acids, which feels slightly out of tune with the broader focus of the introduction. I recommend either replacing these with more general references or incorporating additional citations for balance.

We thank the referee for this hint. As the two references are relatively similar, we have removed Ref 6 and added three more general references to cover a much wider set of compounds:

1. Matsika, S., Modified Nucleobases, *Top. Curr. Chem.* **355**, 209–243 (2014).
2. Marchetti, B., Karsili, T. N. V., Ashfold, M. N. R., Domcke, W., A 'bottom up', ab initio computational approach to understanding fundamental photophysical processes in nitrogen containing heterocycles, DNA bases and base pairs, *Phys. Chem. Chem. Phys.* **18**, 20007–20027 (2016).
3. Schalk, O., Galiana, J., Geng, T., Larsson, T. L., Thomas, R. D., Fdez. Galván, I., Hansson, T., Vacher, M., Competition between ring-puckering and ring-opening excited state reactions exemplified on 5H-furan-2-one and derivatives, *J. Chem. Phys.* **152**, 064301 (2020).

2. Figure 2D is complex and not self-explanatory. I suggest providing a brief explanation in the text to aid readers. Additionally, the term "binding energy" appears several times and would benefit from some contextualization.

We thank the referee for pointing out that flaw. We guess that the referee is referring to Fig. 1D, as Fig. 2 only reaches up to C. In our resubmission, we hope to have described the figure more accurately and put the binding energy better into context.

We have made the following changes to the manuscript (in green):

'Furthermore, we directly reference the change in structural symmetry to the changes in the electronic state of the molecule obtained in a previous study [18]. There, Mayer *et al.* deduced the electronic character from distinct shifts in the X-ray photoelectron spectra, as shown in Fig. 1D. The figure shows a false color representation of the change in the sulfur 2p photoelectron spectra recorded at different delays after UV excitation. To accomplish this, we plot the difference in the S 2p photoelectron spectra with and without optical excitation as a function of the sulfur 2p binding energy and the delay. The data were obtained at the FLASH free-electron facility at much softer x-ray photon energies as described in Ref.[18]. The observed increase in binding energy from point 1 to point 3 resulted from the increase in $^1n\pi^*$ character of the molecule's electronic configuration. This is due to the higher positive local charge at the sulfur atom in the $^1n\pi^*$ state compared to the optically excited $^1\pi\pi^*$ state, which reduces the S 2p core hole screening and thereby increases its binding energy.'

3. Including a graphical representation of the alpha angle in one of the schemes presented would be helpful for readers.

We have changed Fig. 3 and added a sketch showing the definition of angle α as suggested by the referee.

Reviewer #3 (Remarks to the Author):

The manuscript has been professionally, clearly and carefully written and the authors obviously mastered their experimental technique. They also do a good job explaining the background of chemical reactivity, internal conversion and its link to symmetry breaking by nuclear motion. This scientific theme is well established here and shows that the paper is of high contemporary interest and that the out-of-plane distortions are an important aspect of the relaxation mechanisms of a large class of molecules. The analysis of the data has been performed carefully and the quality of the results, in terms of CEI, are very good. The conclusions from the experimental data are well-formulated and show that the paper clearly contributes to our picture of photoinduced dynamics in small quantum systems.

On the other hand, the authors also stress repeatedly that they are using a new experimental method that opens new avenues for research (e.g. "Our experimental scheme opens new avenues for time-resolved studies of complex molecules in the gas phase"). This presumably enhances the novelty factor of their submission. In contrast to the convincingness of their scientific claims, the instrumental/methodological ones are not very convincing. The Coulomb Imaging technique (CEI), while relatively new, has been developed actively during the past years (involving the present authors) and there are several publications devoted to this technique as a means to obtain (also time-resolved) "snapshots" of the structural information of a single molecule. This is by now well-known in the community, and CEI is hardly a novel technique anymore. Also, as supposedly novel details, it is for example already known that the hydrogen ions can be an excellent probe in the CEI of the molecular geometry and that CEI can be successfully coupled with a pump-probe scheme (just as one example, see Crane, S. W., Lee, J. W. L., Ashfold, M. N. R. & Rolles, D. Molecular photodissociation dynamics revealed by Coulomb explosion imaging. *Phys. Chem. Chem. Phys.* 25, 16672–16698 (2023).) Other than that, it is not clear, what exactly is the outstanding experimental and methodological novelty of this study, apart from the normal evolution of a technique.

Of course, the system studied is new, and perhaps more complex than the systems before (as a natural evolution of the method). Also, the particular geometry change -- the planar symmetry breaking -- has perhaps not been observed before by CEI. But it remains unclear, how this constitutes a qualitative important new step, compared to observing any other geometry change. From statements like "While we demonstrate the validity of the time-resolved X-ray CEI approach to molecular symmetry ...", it is not clear, why would molecular symmetry change be such a special circumstance from the viewpoint of CEI?

In this referee's opinion, the authors have overstated their instrumental/methodology novelty claims and should either be more specific about why this study constitutes a very important advancement over what is already established, or tone down these claims. Whether the scientific results warrant publication in *Nature* or in a more specialized high-impact journal, has no obvious answer.

We are thankful for the referee's assessment regarding the overall quality of our study and our paper. With respect to instrumental novelty: To our opinion it is indeed the increased complexity of the molecule and the set of time-resolved observables inspected in our current work, which pose a larger step with respect to technological advancement. The examples

presented in the review article mentioned by the referee are either static cases (i.e. non-time-resolving studies) or target less complex systems. In the vast majority of examples, the information is extracted from up to a three-body breakup of the inspected molecules and/or (for time-resolved studies) from measuring the KER of a fragmentation channel.

While pioneering work by, for example, the Stapelfeldt group targeted already larger molecules, publications where molecular systems of the complexity of 2-thiouracil are inspected "in full" have only recently emerged - mainly with the availability of X-ray free-electron lasers. Our current work is a first experiment from this branch of studies showing time-resolved results. The recent 1-2 years have highlighted the urge to find suitable ways of representing the increasingly complex momentum-space data obtained in such "full" CEI measurements. Again, inspecting the literature, X-ray free-electron laser CEI-work has (as we believe) coined the field here (and e.g. work employing table-top lasers followed now the analysis approaches implemented initially in XFEL CEI-studies). We hope to have provided further ideas on how to extract information (e.g. on molecular symmetry) from such complex time-resolved datasets in our present work. In particular, we showed that an inversion of measured momentum space to position space (which is still an unsolved problem for molecules larger than a few atoms) is not necessary to trace nuclear dynamics along a complex electronic deexcitation route of such a (in CEI-terms) "large" molecule in the time domain.

That being said, we clearly do not want to overstate our claims with respect to novelty. We have changed our manuscript in order to comply with the request by the referee by being more specific thus toning down these statements. We replaced the following sentence in the abstract:

"Our experimental scheme opens new avenues for time-resolved studies of complex molecules in the gas phase."

by a more specific description of why we believe in the impact of our present work:

"We expect that our new analysis approach focusing on these peripheral protons can be adapted as a general concept for future time-resolved studies of complex molecules in the gas phase."

We changed the sentence at the end of the introductory paragraph, which was mentioned by the referee above. The initial aim was not to highlight that CEI "surprisingly" allowed us to observe symmetry properties of a molecule, but to provide an outlook on possible future studies of interest, which could employ our overall approach (XFEL-CEI, peripheral protons as messengers). It now reads:

While we successfully applied the time-resolved X-ray CEI approach to molecular symmetry on one particular heterocyclic molecule, there are no fundamental obstacles that limit its generalization to other heterocycles and even beyond.

Specific comments:

p. 3: "The discussed internal conversion requires an out-of-plane molecular motion belonging to the A" irreducible representation." This was not immediately clear to this reviewer. Is it a

specific vibrational normal mode of A'' or is this connection explained on some other way? This statement would benefit from clarification/illustration.

The sentence mentioned by the referee describes a formal concept: Formally, point group symmetry can only prescribe that some couplings *must* be zero, but it does not make any statement about couplings that are not symmetry-forbidden. Thus, formally any A'' (out-of-plane) vibration is potentially able to induce $A' \rightarrow A''$ transitions. However, in practice there will be modes that couple the states strongly and other modes that have essentially no effect, even though they are out-of-plane. Which concrete modes are most important depends on the system and is not easy to say. Even when theorists run nonadiabatic dynamics simulations, we can not easily pin down *the* one important mode in most cases, although this is a favorite question by experimentalists. Often, a large number of modes provide small to medium contributions. To clarify the formal aspect, we changed the corresponding sentence to:

“The discussed $A' \rightarrow A''$ internal conversion formally requires the activation of at least one out-of-plane molecular motion belonging to the A'' irreducible representation, i.e., internal conversion necessitates some out-of-plane molecular motion.”